# Intracranial Efficacy of Atezolizumab, Bevacizumab, Carboplatin, and Paclitaxel in Real-World Patients with Non-Small-Cell Lung Cancer and EGFR or ALK Alterations

**DOI:** 10.3390/cancers16071249

**Published:** 2024-03-22

**Authors:** Marcus Rathbone, Conor O’Hagan, Helen Wong, Adeel Khan, Timothy Cook, Sarah Rose, Jonathan Heseltine, Carles Escriu

**Affiliations:** 1School of Medicine, University of Liverpool, Liverpool L69 3BX, UK; hlmrathb@liverpool.ac.uk (M.R.); c.a.o-hagan@student.liverpool.ac.uk (C.O.); 2The Clatterbridge Cancer Centre, Liverpool L7 8YA, UK; helen.wong1@nhs.net (H.W.); adeel.khan15@nhs.net (A.K.); tim.cook@nhs.net (T.C.); sarah.rose4@nhs.net (S.R.);

**Keywords:** lung cancer, brain metastases, EGFR, ALK, Atezolizumab, immunotherapy, Paclitaxel, checkpoint inhibitor, PDL1

## Abstract

**Simple Summary:**

Patients with oncogene-addicted lung cancers that have developed resistance to oral targeted therapy have limited treatment options. Several studies have shown conflicting results in this setting, but those using Bevacizumab and Paclitaxel in combination with immunotherapy (ABCP) have consistently shown activity. Spread to the brain is common, but it is not known how many patients with brain metastases are amenable to chemo-immunotherapy, and its effect in this patient group has not been reported to date. We present here our retrospective real-life experience in patients treated with ABCP. Our results in 34 patients underlined that spread to the brain occurred in 19 patients (56%), 17 of them without previous brain radiotherapy. In these 19 patients, chemo-immunotherapy triggered frequent and quick responses inside and outside the brain and showed similar outcomes to those shown in patients with smoking-related lung cancer and previously untreated brain metastases.

**Abstract:**

Contrary to Pemetrexed-containing chemo-immunotherapy studies, Atezolizumab, Bevacizumab, Carboplatin, and Paclitaxel (ABCP) treatment has consistently shown clinical benefit in prospective studies in patients with lung cancer and actionable mutations, where intracranial metastases are common. Here, we aimed to describe the real-life population of patients fit to receive ABCP after targeted therapy and quantify its clinical effect in patients with brain metastases. Patients treated in Cheshire and Merseyside between 2019 and 2022 were identified. Data were collected retrospectively. A total of 34 patients with actionable EGFR or ALK alterations had treatment with a median age of 59 years (range 32–77). The disease control rate was 100% in patients with PDL1 ≥ 1% (*n* = 10). In total, 19 patients (56%) had brain metastases before starting ABCP, 17 (50%) had untreated CNS disease, and 4 (22%) had PDL1 ≥ 1%. The median time to symptom improvement was 12.5 days (range 4–21 days), with 74% intracranial disease control rates and 89.5% synchronous intracranial (IC) and extracranial (EC) responses. IC median Progression Free Survival (mPFS) was 6.48 months, EC mPFS was 10.75 months, and median Overall Survival 11.47 months. ABCP in real-life patients with brain metastases (treated or untreated) was feasible and showed similar efficacy to that described in patients without actionable mutations treated with upfront chemo-immunotherapy.

## 1. Introduction

Lung cancer is the second most common cancer in the world and the most lethal [1]. A subgroup of patients with non-small-cell lung cancers (NSCLCs) has actionable mutations. They are more likely to present at a younger age, with no or light smoking history, Asian ethnicity, adenocarcinoma histology, a higher proportion of PDL1 < 1%, a higher percentage of stage IV disease, and a higher incidence of brain metastases [2,3,4,5]. EGFR mutations and ALK translocations are the best-described actionable mutations. Targeted therapy with small-molecule tyrosine kinase inhibitors (TKIs) is highly effective, but acquired resistance to TKIs invariably occurs [6]. Post-TKI treatment options are limited by co-morbidities and central nervous system (CNS) progression, and are heavily reliant on cytotoxic chemotherapy [7], which has little CNS effect [8].

Immunotherapy has changed the paradigm of lung cancer treatment [9,10]. It offers the possibility of long-term disease control [11,12] and CNS efficacy [13], with a different and overall less toxic profile than chemotherapy [14]. However, tumours with EGFR mutations and ALK translocations showed no obvious benefit in second-line studies with single-agent checkpoint inhibitor therapy [15,16,17]. The presence of these actionable mutations may contribute to resistance to immunotherapy [18,19,20].

Not surprisingly, patients with EGFR mutations or ALK translocations were initially excluded from most first-line checkpoint inhibitor trials [21,22] with the exception of the IMpower150 study [23], where a quadruple combination of Atezolizumab, Bevacizumab, Carboplatin (AUC6), and Paclitaxel (200 mg/m^2^) (ABCP) was tested versus first-line platinum doublet chemotherapy with Atezolizumab or Bevacizumab. Patients were included in the study before or after TKI, and those with untreated brain metastases were excluded. An unplanned subgroup analysis of patients with EGFR mutations reported the outcomes of 34 patients treated with ABCP before or after exhausting oral targeted therapy options versus patients treated with chemotherapy and Bevacizumab (*n* = 45) or Atezolizumab (*n* = 45) [24]. The results suggested that (a) the quadruple paclitaxel-containing chemo-immunotherapy regimen had clinical benefit in patients with actionable mutations, and (b) the benefit was only observed when Bevacizumab and Atezolizumab were given together. Long-term survival with ABCP was later observed in 16 patients with uncommon EGFR mutations in a recent retrospective study from nine German centres [25].

In lung cancer patients who present with brain metastases, pre-immunotherapy evidence suggests that survival is limited to less than 3 months [26], and extracranial disease is consistently a poor prognostic factor [27,28] responsible for over 90% of deaths in this patient group [29]. As discussed before, the incidence of brain metastases in patients with actionable mutations is known to be high [4,5], but the proportion of patients with CNS involvement amenable to ABCP therapy after TKI progression and their outcomes remains unknown.

The evidence of ABCP in patients with ALK-positive lung cancer is limited, but not negligible. A figure in the Appendix A of the IMpower150 study [24] showed a trend towards benefit in the 31 patients with ALK translocations who received ABCP or BCP. The data were not mature enough to show median Overall Survival (mOS) in the ABCP subgroup after more than 19 months of follow-up. Perhaps because of the frequent rapid clinical progression after TKI therapy, chemo-immunotherapy has been poorly studied in patients with ALK translocations in this setting and represent an area of need. There is a lack of randomised prospective evidence of survival in a large study with ALK-positive patients, and now that Bevacizumab is off patent, this is unlikely to occur, whereas new, highly prized antibody drug conjugates are in vogue [30]. The outcomes of ALK-positive patients with or without CNS involvement post-TKI remain unknown. Real-life outcomes are the only available source of data.

The Clatterbridge Cancer Centre caters for the oncology needs of a population of 2.3 million people from seven different NHS Trusts. A centralised consultant-led service was set up in 2015 specifically for lung cancer patients with actionable mutations, with pre-defined imaging protocols that include brain imaging on TKI progression and consistent criteria for follow-up and treatment switch throughout.

High-dose ABCP (as per Impower150) has been reimbursed by the National Health Service in the UK for patients with a wide variety of actionable genomic alterations for several years now. The clinical trial definitions for patients with brain metastases are variable, and this fluidity poses a challenge when adapting treatments to real-life patients. In 2019, aiming to aid treatment individualisation, we interpreted the definition of “symptomatically active brain metastases” as patients with CNS symptoms responsible for clinical deterioration irreversible with dexamethasone 4 mg/day and/or antiepileptic medication. Later on, FDA guidelines [31], intending to maximise patient access to clinical trial recruitment, broadened the definition of active brain metastases to all patients with untreated brain metastases. Immunotherapy CNS efficacy in lung cancer patients has created a still unmet need for specific, consensus-based definitions that could help minimise real-life practice heterogeneity.

In the meantime, this gap has created the possibility of gathering valuable real-life evidence in patients treated with ABCP and a unique set-up to answer valuable questions in the real-life setting, such as the following: What are the demographic characteristics, PDL1 expression, and the incidence of brain metastases (treated or untreated; symptomatic or asymptomatic) of patients fit to receive high-dose chemo-immunotherapy? How often does ABCP toxicity lead to death or treatment discontinuation? What is the CNS effect of ABCP in real-life patients? Does the benefit of translate to a predominantly Caucasian European population? Does the presence of PDL1 expression or EGFR T790M mutation influence responses to ABCP in real-life practice? And is ABCP a viable option in patients with ALK translocations?

Here, we aimed to identify the proportion of our patients with EGFR mutations or ALK translocations who were fit for ABCP post-TKI exhaustion and quantify their outcomes.

## 2. Materials and Methods

### 2.1. Patient Selection and Management Criteria

Because of our centralised consultant-led set-up, the treatment and follow up criteria were consistent throughout. Patients were offered ABCP after exhausting all TKI options regardless of their tumour genetic anomaly. Because of the high-dose chemotherapy of the ABCP regimen, patients were prospectively selected to have no significant co-morbidities, no contraindications to immunotherapy (namely, interstitial lung disease or ongoing TKI-induced pneumonitis) or to VEGF inhibition (no recent symptomatic pulmonary embolism and no tumour invading central vessels), controlled CNS symptoms defined as responsive to up to 4 mg/day of dexamethasone and/or antiepileptic medication, and acceptable WHO performance status. The first symptom and toxicity assessment was performed before cycle 2. Imaging was performed during cycle 3 (after 9 weeks of treatment) and every 3 months thereafter unless there was a clinical need to scan before. Brain imaging was performed at baseline and with every radiological assessment if intracranial spread was identified before ABCP treatment; otherwise, repeat brain imaging was performed on a yearly basis or if CNS symptoms occurred before the scheduled imaging. Data analysis was performed in April 2023.

### 2.2. Definitions

In the response assessment, progressive disease was defined as radiological tumour growth with or without clinical deterioration. Stable disease was defined as minor changes in tumour size or mixed response with no clinical deterioration. Partial response was defined as obvious volume reduction with or without clinical improvement. Progression-Free Survival (PFS) was defined as the time from day 1, cycle 1 of ABCP until radiological or clinical progression, last contact with the patient, or death of any cause. Central nervous system (CNS) progression amenable to stereotactic radiotherapy was considered oligo-progression and maintenance treatment was continued (this scenario would limit the CNS PFS, but not the overall PFS). Radiological response was defined as tumour volume reduction of any magnitude. Clinically meaningful improvement or resolution of stable CNS symptoms pre-treatment was considered symptomatic response. Grade 3–4 events were defined as those triggering dose reduction, treatment delay, or discontinuation with or without hospital admission. Derived Neutrophil Lymphocyte Ratio (dNLR) was calculated as the neutrophil count divided by the lymphocyte count, minus the lymphocyte count.

### 2.3. Statistical Methods

Audit committee approval was obtained in our institution before proceeding with data collection in accordance with the Declaration of Helsinki. Data were collected retrospectively, and a radiology review was performed on two different occasions by different authors.

All patients treated with ABCP between 2019 and December 2022 were selected. Univariate Cox regression analysis was used to analyse the entire population and Pearson’s correlation for the CNS population. Overall Survival (OS) was defined as the time from day 1, cycle 1 of ABCP until death or the date of last contact with the patient. One month equalled 30.42 days. Hazard ratios for OS and PFS were calculated using the log rank test, and *p*-values using the Gehan–Breslow–Wilcoxon test. No patients were lost to follow-up. Data were presented for all patients, those with brain metastases, and those with ALK-positive disease to maximise patient group homogeneity and minimise bias.

This manuscript was written in accordance with the STROBE guidelines.

## 3. Results

### 3.1. Patient Characteristics

A total of 34 patients were treated with ABCP between March 2019 and August 2022. Their average age was 59 and there was a slightly higher proportion of women (56%) (see Table 1). The proportion of NSCLC Not-Otherwise-Specified (NOS) was 41%. The rest of the patients had adenocarcinoma. All patients had stage IV disease; seven patients (21%) had liver metastases, but the most common metastatic burden in these patients was in the brain. Nineteen patients (56%) had brain metastases, seven of them (21%) with well-controlled symptoms. Ten patients (29%) did not have enough tissue for PDL1 testing, but amongst those that could be tested, PDL1 expression was low. In addition, 14 out of 24 patients (58.33%) had PDL1 < 1%, and 4 (17%) had PDL1 ≥ 50%, while 29 patients (85%) had EGFR-mutated cancers, most of them (74%) with common mutations. Five patients had ALK+ lung cancer, four of them with brain metastases. Of the 19 patients with brain involvement, 15 (79%) had EGFR tumours, and 4 had received previous brain radiotherapy.

### 3.2. Efficacy and Tolerability

At the time of data analysis, 8 out of 34 patients (23.5%) were alive, 3 patients had completed 2 years of treatment, and 4 patients were receiving maintenance treatment. The median treatment of ABCP for the 34 patients was seven cycles (see Table 2).

Disease control was achieved in 25 patients (74%). Of the 11 patients with cancer-related symptoms, the median time to meaningful symptom improvement or resolution was 12 days. The median Progression-Free Survival (mPFS) for the entire group (*n* = 34) was 6.71 months and the median Overall Survival (mOS) was 8.15 months (see Table 2 and Figure 1).

A univariate Cox regression analysis of the overall population did not identify any subgroups that correlated with OS with statistical significance, but there was a trend towards improved outcomes in patients with female sex (HR 0.38; 95%CI 0.033–3.04), ALK+ disease (HR 0.55; 95%CI 0.001–56.50), or the presence of CNS disease (HR 0.21; 95%CI 0.006–2.15).

The 10 patients with unknown PDL1 results showed an even distribution amongst the radiological response group (see Appendix A), with 30% of patients having disease progression, 30% stable disease, and 30% a partial response. Amongst the 24 patients with PDL1 results, 43% (*n* = 6/14) of those with PDL1 < 1% had disease progression, whereas 80% (*n* = 8/10) of those with PDL1 ≥ 1% had a partial response. All patients with PDL1 ≥ 1% achieved disease control, and the presence of PDL1 expression had a positive correlation with radiological response (r = 0.4191 (95%CI 0.01895–0.7036); r^2^ = 0.1757; *p* = 0.0415), but the low r suggested that the absence of PDL1 expression did not guarantee a lack of response. There was no correlation between the presence of PDL1 expression and PFS or OS, but there was a trend towards better survival observed in patients with PDL1 ≥ 1% (HR 0.44; 95%CI 0.18–1.1; *p* = 0.0896) (see Figure 2B).

When analysing patients with common EGFR mutations (*n* = 29), the mPFS was 5.39 months and the mOS was 7.36 months. There were no differences between patients with L858R (*n* = 8) and deletion 19 (*n* = 17) mutations either for PFS (HR 0.82; 95%CI 0.33–2.06; *p* = 0.72) or for OS (HR 1.67; 95%CI 0.72–4; *p* = 0.1772). The mPFS was 5.39 months and the mOS was 7.13 months for patients with deletion 19. For those with the L858R mutation, the mPFS was 5.6 months and the mOS was 8.89 months.

Regardless of the original background mutation, as can be seen in Figure 2C,D, the presence of a T790M-acquired resistance mutation (*n* = 8) showed a trend towards worse survival (HR 1.92; 95%CI 0.67–5.5), although with higher disease control rates (87.5% in T790M+ versus 63.16% in T790M−).

Toxicity leading to treatment dose reduction, omission, or discontinuation occurred in 18 patients (53%), 50% of which were chemotherapy-related (see Table 2). Both symptom improvement and toxicity were most common during the first treatment cycle. No toxicity led to complete treatment discontinuation, but it did lead to single-drug discontinuation: two patients stopped Paclitaxel due to allergic reaction and one patient stopped Bevacizumab after a pulmonary embolism. For most patients, chemotherapy toxicity resolved after a dose reduction or single-drug discontinuation, and all patients who had immunotherapy interrupted due to toxicity resumed treatment after a course of appropriate immunosuppressive therapy. Compared with the overall population, there were no differences in the tolerability signals in patients with brain metastases or those with ALK+ disease.

### 3.3. Patients with CNS Involvement

A total of 19 patients had brain metastases when starting treatment with ABCP chemo-immunotherapy after TKI exhaustion, 8 (42%) had PDL1 < 1%, 7 (36.8%) had well-controlled CNS symptoms on dexamethasone and/or antiepileptic medication, and 15 (79%) had untreated brain disease. Of the two patients with previously treated brain disease, one had a growing metastatic deposit near the fourth ventricle in a previously irradiated site, and one who had been treated previously with whole-brain radiotherapy would have been qualified as having active brain metastases according to FDA guidelines [31]. The average time to intracranial symptom improvement or resolution was 12.5 days, always within the first treatment cycle (range 4–21 days). One patient (5%) had intracranial complete radiological response and seven (37%) had a partial response, with an overall 74% intracranial disease control rate. The intracranial and extracranial responses were mostly synchronous (89.5%). As can be seen in Figure 3, the intracranial (IC) mPFS was 6.48 months, lower than the extracranial (EC) mPFS of 10.75 months, with an mOS of 11.47 months, which was higher than the 8.15 months of the overall population.

In Figure 4, we plotted the length of time on TKI therapy for each patient on the left, with individual outcomes for ABCP on the right. The length of TKI treatment correlated with ABCP clinical outcomes using Pearson correlation: IC PFS (r = 0.7; 95%CI 0.365–0.877, *p* (two-tailed) = 0.0008), EC PFS (r = 0.65; 95%CI 0.279–0.853, *p* = 0.0025) and OS (r = 0.5779; 95%CI 0.168–0.817, *p* = 0.0008).

Intracranial progression was treated with brain radiotherapy while systemic therapy continued. This limited the IC PFS, but not the EC PFS. No new toxicity signals were observed when IC radiotherapy was administered. This was evident in patients 14 and 18, as shown in Figure 4. Amongst the patients who did not have a radiological response, patient 6 in Figure 4 had an intracranial response and extracranial progression with superior vena cava obstruction (SVCO) amenable to radiotherapy treatment that remained symptomatic after treatment. Systemic treatment was stopped, but the patient went on to live for another year (11.47 months). Surprisingly, however, amongst the patients who showed a radiological response, the IC mPFS was 16.27 months, higher than the EC mPFS of 14.37 months, but when comparing both populations (*n* = 13 in each group), there was no clear trend or difference between them (HR 1.026, 95%CI 0.38–2.73).

Patients 7, 9, 12, and 18 in Figure 4 had stable disease on CT, suggesting that lack of disease volume reduction may not correlate with the magnitude of the outcome.

As discussed before, all patients with positive PDL1 disease showed a response. Hence, we explored blood prognostic scores and biochemical markers [32] that could predict poor outcomes amongst patients with CNS involvement and PDL1 < 1% (*n* = 8), as this could help avoid ineffective treatment in a subgroup of patients who, due to the presence of CNS disease, would have a poor prognosis, as shown in Figure 4. Only the derived Neutrophil Lymphocyte Ratio correlated with IC PFS (r = 0.73, *p* = 0.039), EC PFS (r = 0.62, *p* = 0.101), and OS (r = 0.72, *p* = 0.046) (see Appendix A). This correlation was not apparent amongst the entire group of patients with CNS involvement regardless of PDL1. Five out of these eight patients showed progression. Of two patients with a dNLR of 2.4, one suffered disease progression and the other exhibited disease response. The rest of the patients who did not respond had a dNLR of <2.4.

### 3.4. ALK+ Lung Cancer

Patients with ALK translocation (*n* = 5) had an mPFS of 18.54 months and an mOS of 19.89 months. When compared with patients with EGFR mutations, those with ALK showed a trend towards better survival (HR 0.37; 95%CI 0.16–0.9; *p* = 0.1706).

Amongst these patients, there was a broad difference between patients who did not benefit and those who did (see Appendix A). Lack of response was evident during the first cycle, whereas two patients who responded continued to benefit and were alive beyond the 2 years of treatment at the time of data analysis.

## 4. Discussion

To our knowledge, this was the largest study to date involving real-life patients with a variety of actionable mutations treated with ABCP combination therapy, and the only one to report specific chemo-immunotherapy outcomes in patients with CNS metastases from EGFR-mutated or ALK-positive tumours after TKI exhaustion. The response and disease control rates were high, and the survival benefit was most evident in patients with ALK translocations and/or brain metastases. Half of our patients, those with highest need for a CNS-active therapy and best outcomes within the group treated with ABCP, would have been excluded from most chemo-immunotherapy prospective studies [33,34,35].

The retrospective nature of the study was an important limitation, restricting the accurate quantification of follow-up and tolerability. Regional characteristics should also be considered when extrapolating our results (we provide oncology care to a large area with poor socio-economic status, which may, itself, shorten patient survival). As expected in this low-frequency disease, patient numbers were limited, although large when compared with other published real-life studies in this patient population [25]. On the other hand, the generalisability of our results was supported by the fact that the patients were treated by the same team, which provided an unusual homogeneity of patient selection and care; moreover, the real-life nature of the study ensured patients did not need to go through a screening period that would have excluded rapidly growing tumours. Furthermore, a real-life study was the only option available to assess patients with untreated brain metastases in Europe, and the only way to contextualise the degree of patient selection of prospective studies that tested ABCP [23,24,36].

Recent chemo-immunotherapy studies have shown heterogeneous results in patients with actionable mutations [24,33,34,35,36]. On the one hand, randomised studies with Pemetrexed-containing combinations [33,34,35] showed disappointing results; on the other hand, studies using Paclitaxel and Bevacizumab [24,36] consistently showed therapeutic efficacy, especially in patients with brain metastases, PDL1 ≥ 1%, and/or ALK translocated lung cancers. This is consistent with our results.

Despite these data, the value of immunotherapy in patients with EGFR mutations is currently being questioned [37]. In chemo-immunotherapy studies with Pemetrexed instead of Paclitaxel with Bevacizumab [33,34,35], the mOS of the interventional arms ranged between 15.9 and 20.7 months, and in the control arm between 14.7 and 18.7 months. In the pre-immunotherapy era, Pemetrexed showed improved survival in non-squamous patients [38]. In the PARAMOUNT study [39], patients who had maintenance Pemetrexed had an mOS of 13.9 months, and a later systematic review of the clinical trial population [40] showed that patients with EGFR mutations (without untreated brain metastases) who received Pemetrexed-containing regimens after TKI exhaustion had a weighted mOS of 15.91 months. Therefore, most novel chemo-immunotherapy studies chose to disregard the possible immunotherapy-potentiating effect of Paclitaxel [41,42,43] in favour of the possible cytotoxic advantage of Pemetrexed chemotherapy. IMpower151 [35] claimed to use the original ABCP regimen, but 97.4% of patients received Pemetrexed instead of Paclitaxel, although, in contraposition with the other studies, it included Bevacizumab in both arms and enrolled patients with EGFR and ALK genomic alterations (as well as patients without actionable mutations).

The only study to use the original Bevacizumab- and Paclitaxel-containing ABCP from IMpower150 was the Korean ATTLAS study [36], albeit with lower chemotherapy doses (Carboplatin AUC5 and Paclitaxel 175 mg/m^2^). Over 40% of the patients had brain metastases. The ORR was 69.5% in the ABCP arm vs. 41.9% in the control arm, and the respective mPFS was 8.48 months vs. 5.62 months; HR 0.62 (95%CI 0.45–0.86). The primary end point was PFS with a 2:1 randomisation, and the study did not demonstrate a survival benefit that was not powered to quantify. Surprisingly, the mOS was 20.27 months for the control group, which was substantially longer than expected [40], especially when considering the large proportion of patients with brain metastases, which brings into question the degree of CNS involvement in the patient population included and/or the possibility of crossover on subsequent treatment lines. Accounting for the lower numbers in the control arm inherent to the 2:1 randomisation, the subgroup analysis suggested that the benefit was driven by patients with PDL1 ≥ 1%, CNS involvement, and/or an acquired T790M EGFR resistance mutation.

### 4.1. PDL1 Was a Positive Biomarker of Response

The large proportion of patients with NSCLC Not-Otherwise-Specified (NOS) or with unknown PDL1 status is due to limited diagnostic tissue availability [44,45], which is to be expected in real-life practice. During the diagnostic pathway, in patients with no smoking history, genomic profiling is prioritised, which may hinder tumour subtyping or less relevant marker testing such as PDL1; never-smokers are offered chemo-immunotherapy regardless of the PDL1 levels [46]. Amongst the 24 patients with PDL1 results pre-TKI therapy, the largest group was the one with PDL1 < 1%, with 14 patients (58.3%). There was also a high incidence of CNS metastases (56%). These features were consistent with previous descriptions of similar patient populations [2,4].

Amongst our patient group with multiple mutation types and available PDL1 results, PDL1 positivity at diagnosis showed a trend towards better outcomes (Figure 2A,B). Our results are consistent with the IMpower150 study [24] and the ATTLAS study data [36], where PDL1 positivity correlated with better OS.

In our patient cohort, all patients with positive PDL1 results showed radiological response—in patients with actionable mutations, PDL1 ≥ 1% may be a positive predictive marker of response to chemo-immunotherapy. The necessary adjustments in the diagnostic pathway or repeat biopsies for PDL1 testing on TKI progression may help inform treatment decisions in this setting.

### 4.2. Patients with EGFR-Mutated Tumours

Amongst our patients with EGFR mutations (*n* = 29), the mPFS and mOS were 5.39 and 7.36 months, respectively. In the IMpower150 study [24], there were 26 patients with common mutations presenting after TKI exhaustion treated with ABCP. The outcomes of the intention-to-treat population (*n* = 34), which included patients treated before TKIs and patients with rare EGFR mutations, were substantially better than in our study: the OS at 24 months was 65% and the mOS was not reached. In the ATTLAS study [36], the mPFS was 8.48 months and mOS 20.63 months. This very significant quantitative difference in outcomes could have been related to (a) patient selection due to the real-life nature of our study and with rapidly growing disease that would not have been able to go through trial screening, (b) the TKI survival impact of post-ABCP therapy in the IMpower150 group, (c) a regional bias from our single-centre study, and (d) a low patient number bias in our study.

In our patient cohort, patients with a T790M-acquired resistance mutation showed a higher rate of radiological response, but worse survival outcomes (Figure 2C,D). This was consistent with the results from the ILLUMINATE study presented at the World Lung Conference 2023 [47]. In that study, all patients had been treated with Durvalumab, Tremelimumab, and platinum–Pemetrexed chemotherapy, but recruited in two parallel arms according to the presence or absence of a T790M mutation. Patients with a T790M mutation showed lower response rates and disease control rates than those without the mutation, but the study was not randomised and the populations may have been different. The IMpower150 study [24] only had three patients with T790M mutations, and we had eight patients in our study. Number bias could explain why our cohort of patients with T790M+ disease had higher response rates, but it questions the possibility that T790M may be a predictive factor of response. It may only be a biomarker of poor prognosis. In any case, T790M has limited relevance now in patients treated with first-line Osimertinib [48], the current standard of care.

### 4.3. Patients with CNS Involvement

In our patient population, brain metastases were common (56%), and the presence of CNS disease showed a trend towards improved outcomes in Cox regression analysis. In our study, 7 out of the 19 patients (37%) were on steroids and/or antiepileptic medication, but the number of patients with active brain metastases was higher if we use the very broad FDA guideline definition “where patients have new brain metastases or progressive brain metastases that have not been subjected to CNS-directed therapy since documented progression” [31]. Moreover, 17 out of our 19 patients had untreated brain metastases and would have been excluded from prospective studies [24,35,47,49,50].

The ATEZO-BRAIN trial [51] (*n* = 40 patients without actionable mutations and brain metastases, treated with upfront systemic Atezolizumab, Carboplatin, and Pemetrexed) showed intracranial responses of 42.7% according to RANO criteria, with 90% synchronous responses between intracranial and extracranial disease and IC mPFS of 6.9 months. In our study, the response rates were 42%, intra- and extracranial synchronicity 89.5%, and IC mPFS 6.48 months, suggesting that CNS efficacy with ABCP in patients with actionable mutations may be similar to that described in patients treated with chemo-immunotherapy without actionable mutations. The fact that patients with brain metastases in our study and the ATEZO-BRAIN trial [51] showed both high responses with initial CNS control and delayed extracranial progression supports further prospective randomised studies comparing upfront brain radiotherapy versus upfront chemo-immunotherapy in lung cancer patients regardless of the presence of actionable mutations.

Interestingly, previous length of TKI treatment correlated with IC PFS, EC PFS, and OS in patients with CNS disease (see Figure 4). Translational studies have shown that TKI therapy may sensitise patients to immunotherapy by maximising antigen presentation [52] and modifying the extracellular matrix to facilitate immune infiltration [53]. Our results support these data in real-life patients in a length-of-treatment-dependent manner.

Amongst our patient population, upfront systemic chemo-immunotherapy had an intracranial disease control rate of 74% (*n* = 14) (see Table 2), and 79% of patients (*n* = 16) lived more than 3 months (two patients without radiological response per se lived 4.14 and 11.47 months). Not surprisingly, intracranial responses were key in obtaining a short-term benefit. However, in the longer term, once patients had been exposed to ABCP, the extracranial disease appeared to be less relevant in influencing survival. In patients who showed a response to ABCP, the IC mPFS (16.27 months) was closer to the mOS (17.92 months) than the EC mPFS (14.32 months), and patient 6 in Figure 4 exhibited intracranial response and extracranial progression but went on to live 11.47 months. CNS radiotherapy during immunotherapy treatment has shown no increased toxicity in lung cancer patients [54,55]. If the brain disease is indeed responsible for survival once patients have been exposed to ABCP, we can posit here that sequencing radiotherapy to the residual brain disease may have a positive survival impact. Prospective studies are needed to explore this further.

We also actively looked for negative predictive factors of response. PDL1 positivity appeared to be a guarantee of benefit, but further ways of anticipating progression amongst patients with PDL1 < 1% could help avoid ineffective treatment in patients with poor prognosis. Amongst patients with CNS involvement and PDL1 < 1%, dNLR correlated with PFS and OS, and all patients who did not benefit from ABCP had a dNLR < 2.4. Further prospective confirmation and exploration of other negative biomarkers of response to immunotherapy in this patient population are needed.

### 4.4. Patients with ALK-Translocated Tumours

Symptomatic brain metastases in patients with ALK translocations are common after TKI progression. In the patient group described here, four out of five patients had brain metastases. In our clinical experience, before we started using chemo-immunotherapy, consistently with the reported outcomes of lung cancer patients with brain metastases [26], disease progression post-TKI is fast and the prognosis is less than 3 months with or without treatment; this is a population of high need, and, in our real-life daily practice, the benefit of ABCP was most noticeable here.

There is, however, a reticence to use chemo-immunotherapy in patients with ALK translocations. Several reasons may account for this. These patients with low-incidence cancers do exceptionally well on TKI treatment and any further benefit post-TKI may be perceived as irrelevant and much more toxic in comparison.

It has also been recommended that Pemetrexed be used in patients with ALK-translocated lung cancer based on first-line, pre-TKI context evidence. Two retrospective studies with 15 [56] and 19 [57] patients treated with platinum and Pemetrexed combinations before TKI in 2011 reported an mPFS of 9.2 and 9 months, respectively, much longer than in patients with EGFR or wild-type lung cancer, thus suggesting that ALK-positive tumours could be highly sensitive to Pemetrexed chemotherapy. Later, larger retrospective and prospective first-line studies of between 70 and 187 patients [58,59,60] suggested that the benefit of Pemetrexed was similar in patients with no smoking history regardless of the presence of an ALK translocation, and reported an mPFS between 7 and 8.5 months. Still, the perception remained that Pemetrexed should be used when treating patients with ALK-translocated tumours.

The presence of ALK translocation is perceived as an absolute contra-indication to have immunotherapy. In 2020, Lin et al. [61] published a retrospective multicentre study with 58 patients treated with Pemetrexed-based combinations, which included small-molecule therapy, Bevacizumab, or immunotherapy, but not Paclitaxel, Bevacizumab, and immunotherapy together as with ABCP. The overall response rates were 29.7%, the CNS response rates were 15.8%, and the mPFS was 4.3 months. Another retrospective study from 2021 with 83 patients diagnosed with ALK-positive lung cancer mostly treated with single-agent immunotherapy described a median time to treatment discontinuation of 2.17 months [16], an outcome that is consistent with our pre-ABCP experience.

The use of Paclitaxel instead of Pemetrexed, together with the long infusion times, the poor outcomes reported with single-agent or Pemetrexed-combination immunotherapy, and the high doses of chemotherapy in the original ABCP regimen, are arguments used for not treating patients with ALK-translocated disease with it.

However, as reported in the Appendix A of the IMpower150 study [24], amongst the 31 patients with ALK-translocated disease treated with either ABCP or BCP, there was a trend towards benefit with ABCP (HR OS 0.47; 95%CI 0.15–1.48); BCP-treated patients had an mOS of 6.9 months, and the mOS for ABCP was not reached after a median follow up of 19.6 months (patients with untreated brain metastases were excluded in this study). ABCP appeared to be different to other immunotherapy combinations. These findings were consistent with our real-life results, where, despite the high incidence of brain metastases (80%), the mOS for patients with ALK-positive disease was 18.31 months.

### 4.5. Paclitaxel versus Pemetrexed

Of all of the chemo-immunotherapy prospective studies in patients with actionable mutations reported so far, the IMpower150 subgroup [24] and the ATTLAS study [36] are the only ones that have shown benefit, and this was consistent with the outcomes observed in patients with rare EGFR mutations [25] and in our study. What they all have in common is the concurrent use of Bevacizumab and Paclitaxel together with platinum and Atezolizumab. All other reported prospective studies [35,47,49] used Pemetrexed instead of Paclitaxel, which has shown in in vitro and in vivo studies that it may have pleiotropic immune-modulating effects by promoting dendritic cell maturation and enhancing proinflammatory cytokine secretion [41,42,43]. If further prospective chemo-immunotherapy studies are considered in patients with actionable mutations, prospective and retrospective evidence supports the use of Paclitaxel with Bevacizumab and not Pemetrexed.

### 4.6. Tolerability

In our study, the rate of grade 3–4 adverse events leading to temporary or permanent treatment discontinuation was 53% (*n* = 18), half of those 26.5% (*n* = 9) due to immunotherapy side effects, but only 3 patients (8%) permanently discontinued one of the treatments due to adverse events. In the IMpower150 study [24], the rates of grade 3–4 adverse events were 63.6% and the rates of any treatment discontinuation were 33.3% in the ABCP arm. Our patient population had a similar tolerability profile; however, in our patient group, chemotherapy dose reductions or single-drug discontinuations occurred in 26.5% of patients. ABCP, as per IMpower150, contains high-dose chemotherapy with Carboplatin AUC6 and Paclitaxel 200 mg/m^2^. This may be unnecessary. This was evident in the ATTLAS study [36], where standard chemotherapy doses were used (Carboplatin AUC5 and Paclitaxel 175 mg/m^2^), and only 7.9% of patients had grade 3–4 adverse events. The frequency of dose reductions and the rapid speed of clinical response in our patient group suggest that a reduced chemotherapy ABCP regimen may render the regimen suitable to a broader patient population and improve its tolerability, whilst still triggering an effective immunotherapy response.

### 4.7. Clinical Implications and Future Directions

Personalised therapy requires effective therapeutic options. In the post-TKI context, and especially in PDL1 ≥ 1% disease and/or the presence of CNS involvement, our results add to the prospective evidence that suggest that ABCP (with Paclitaxel) should be considered.

The suggestion that PDL1 positivity leads to response in a patient population where limited diagnostic tissue is common indicates that repeat biopsies may help guide treatment choice.

Further studies should focus on three key issues. First, there is a need to maximise the tolerability of ABCP. Immediately accessible options would be using lower chemotherapy doses (as per the ATTLAS study [36]) or a reduced chemotherapy cycle number, as in the Checkmate 9LA study [22]. A more complex alternative could consist of a VEGF-inhibiting antibody drug conjugate with a microtubule inhibitor load, in combination with PD1 or PDL1 inhibitors (hence, two drugs instead of four). This may not only reduce adverse effects and increase efficacy, but also reduce the infusion times, which may be appealing in the context of service delivery constraints.

A second focus of research should be on identifying patients with primary resistance to immunotherapy, especially in the context of brain metastases, as this may prevent the unnecessary toxicity of standard therapy and offer the chance to recruit these patients into alternative studies. We have shown here that dNLR has the potential to identify primary resistance in patients with brain metastases and PDL1 < 1%, but this needs prospective confirmation in a larger cohort. We have also shown here that patients with untreated brain disease are common in this therapeutic context, and further studies should target this patient population, especially if they are known to be predicted to have a poor response to systemic therapy; the improvement in a short outcome measure is likely to identify active therapies with minimal investment in a patient population of high need.

Finally, prospective tailored evidence of chemo-immunotherapy is needed in patients with ALK translocations post-TKI progression. These tend to be young, fit patients with a high incidence of CNS involvement and limited standard-of-care options that involve brain radiotherapy and non-CNS active systemic chemotherapy. As discussed above, the perception that these patients should have Pemetrexed should be at least questioned when there is evidence of a systemic CNS-active therapy.

## 5. Conclusions

Our results showed that selected patients with lung cancer brain metastases and EGFR mutations or ALK translocations derived clinical benefit from Atezolizumab, Bevacizumab, Carboplatin, and Paclitaxel treatment despite high toxicity rates. PDL1 positivity and length of previous TKI therapy correlated with immunotherapy benefit. In patients with CNS involvement and PDL1 < 1%, dNLR may help identify patients with primary resistance to immunotherapy that could avoid unnecessary toxicity.

Our real-life data contextualise the results of prospective trials and underline the need to use Bevacizumab and Paclitaxel in combination with immunotherapy when treating patients with actionable mutations. The data highlight the need to focus further immunotherapy prospective trials on the population of need, such as in patients with untreated brain metastases and/or ALK translocations, and not exclusively on patients with the best outcomes when untreated.

## Figures and Tables

**Figure 1 cancers-16-01249-f001:**
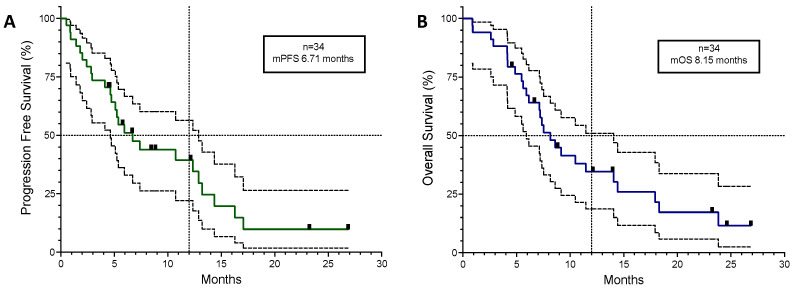
Kaplan–Meyer curves for all 34 patients plotted with 95% confidence intervals (dotted lines). (**A**) Progression-Free Survival. (**B**) Overall Survival. Black squares in the coloured lines represent censored patients. mPFS; median Progression-Free Survival. mOS; median Overall Survival.

**Figure 2 cancers-16-01249-f002:**
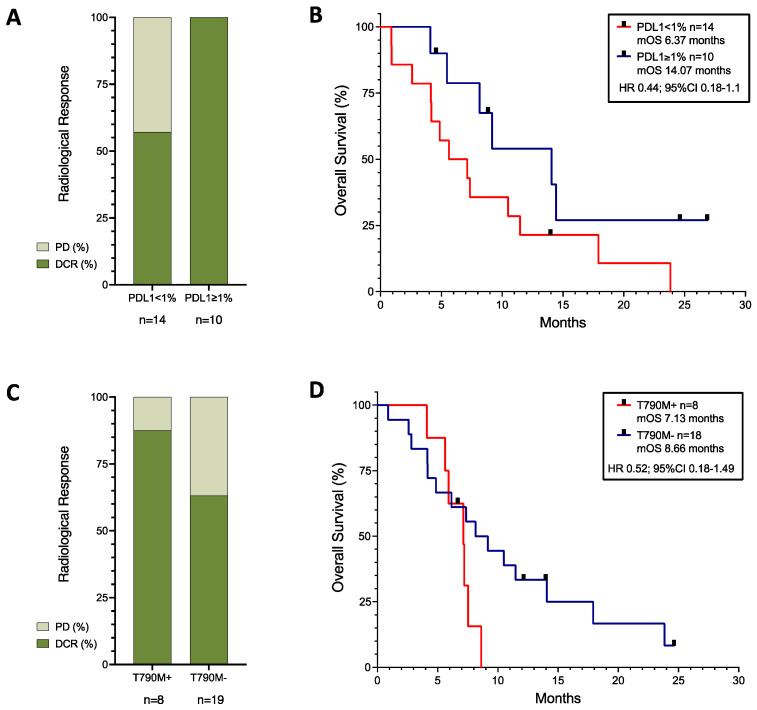
Response rates (**A**,**C**) and survival curves (**B**,**D**) of patients with known PDL1 expression (**A**,**B**), and of patients with EGFR mutation with or without T790M-acquired mutation (**C**,**D**). Black squares represent censored patients. PD; progressive disease, DCR; disease control rate, mOS; median Overall Survival.

**Figure 3 cancers-16-01249-f003:**
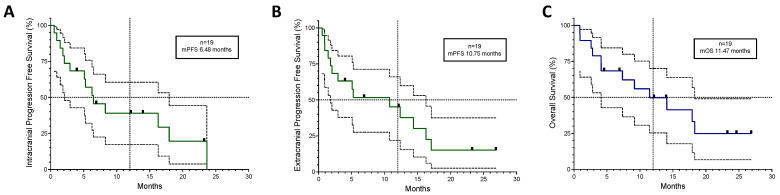
Kaplan–Meyer curves for the 19 patients who had brain metastases before starting ABCP treatment plotted with 95% confidence intervals. (**A**) Intracranial Progression-Free Survival, (**B**) Extracranial Progression-Free Survival, and (**C**) Overall Survival. Black squares represent censored patients. Discontinuous lines mark 50% on the Y axis and the 12-month mark on the X axis. mPFS; median Progression-Free Survival. mOS; median Overall Survival.

**Figure 4 cancers-16-01249-f004:**
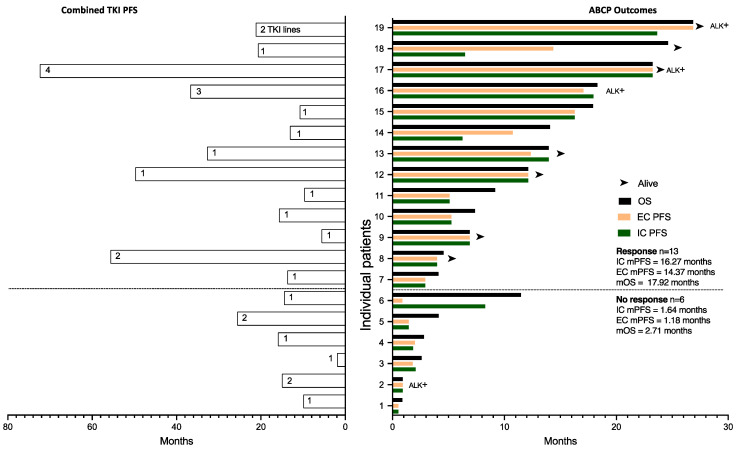
Clinical outcomes of CNS patients plotted for individual patients and separated along the Y axis according to radiological response to ABCP. Combined PFS of previous TKI treatments are plotted on the left for each patient, with the number of TKI treatment lines within each bar. ABCP outcomes (OS from day 1 ABCP, extracranial PFS, and intracranial PFS) are plotted on the right. Patients with ALK translocations are marked as ALK+. mOS; median Overall Survival, mPFS; median Progression-Free Survival, IC; intracranial, EC; extracranial.

**Table 1 cancers-16-01249-t001:** Patient characteristics.

	All Patients	Brain Metastases	ALK+
**Patient number (*n*)**	34	19	5
**Average age (range)**	59 (32–77)	59 (32–77)	55 (35–69)
**Sex *n* (%)**			
Male	15 (44)	9 (43)	2 (40)
Female	19 (56)	10 (47)	3 (60)
**Histology *n* (%)**			
Adenocarcinoma	20 (59)	11 (58)	4 (80)
NSCLC NOS	14 (41)	8 (42)	1 (20)
**Metastatic burden**			
Average number of organs involved (median)	2.7 (3)	3 (3)	2.8 (3)
Liver *n* (%)	7 (21)	4 (21)	1 (20)
CNS *n* (%)	19 (56)	19 (100)	4 (80)
Patients on dexamethasone *n* (%)	7 (21)	7 (37)	0 (0)
**PDL1**			
NK *n* (%)	10 (29)	5 (26)	2 (40)
<1% *n* (%)	14 (41)	8 (42)	1 (20)
1–49% *n* (%)	6 (18)	2 (11)	1 (20)
≥50% *n* (%)	4 (12)	2 (11)	1 (20)
**Genetic profile**			
EGFR	29 (85)	15 (79)	-
EGFR (del19 or L858R) *n* (%)	25 (74)	14 (74)	-
EGFR T790M *n* (%)	8 (24)	2 (11)	-
EGFR rare mutations responsive to TKI *n* (%)	1 (3)	1 (5)	-
EGFR exon 20 ins *n* (%)	3 (9)	1 (5)	-
ALK+ *n* (%)	5 (15)	4 (21)	5 (100)
**Previous treatment**			
Average TKI treatment lines (range)	1.56 (1–4)	1.47 (1–4)	2.6 (2–4)
WBRT *n* (%)	4 (12)	4 (21)	1 (20)

**Table 2 cancers-16-01249-t002:** Outcomes and toxicity. mPFS; median Progression-Free Survival. mOS; median Overall Survival.

	All Patients (*n* = 34)	Brain Metastases (*n* = 19)		ALK+ (*n* = 5)
**ABCP treatment**				
Median cycles (range)	7 (1–35)	7 (1–35)		23 (2–35)
Completed four cycles of chemotherapy *n* (%)	27 (79)	13 (64)		3 (60)
**Radiological response (%)**		**Extracranial**	**Intracranial**	
Complete response	1 (3)	1 (5)	1 (5)	1 (20)
Partial response	18 (53)	8 (42)	7 (37)	3 (60)
Stable disease	6 (18)	4 (21)	6 (32)	0
Disease progression	9 (26)	6 (32)	5 (26)	1 (20)
**Clinical response**		**Extracranial**	**Intracranial**	
Median time to symptomatic response Days (range)	*n* = 11 12 (5–20)	*n* = 7 12.5 (4–21)		*n* = 1 20
12-month PFS (%)	32	48		80
12-month OS (%)	34	52		80
mPFS (months)	6.71	10.75	6.48	17.06
mOS (months)	8.15	11.47		18.31
**Tolerability**				
Grade 3/4 events (%)	18 (53)	9 (47)		3 (60)
Chemo-related or Bevacizumab-related G3/4 events (%)	9 (50)	5 (55)		2 (67)
Immunotherapy-related G3/4 events (%)	9 (50)	4 (45)		1 (33)
Median time to toxicity Days (range)	18 (0–70)	21 (5–70)		14 (10–16)

## Data Availability

The datasets generated during and/or analysed during the current study are not publicly available to guarantee patient anonymisation but are available from the corresponding author on reasonable request.

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
