# Peer review of "Intracranial Efficacy of Atezolizumab, Bevacizumab, Carboplatin, and Paclitaxel in Real-World Patients with Non-Small-Cell Lung Cancer and EGFR or ALK Alterations"

_cancers, 2024, doi:10.3390/cancers16071249_

Round 1

Reviewer 1 Report

Comments and Suggestions for Authors

The study conducted by the authors between 2019 and 2022 evaluated the clinical effectiveness of Atezolizumab, Bevacizumab, Carboplatin, and Paclitaxel (ABCP) in lung cancer patients with actionable EGFR or ALK alterations, focusing on those with brain metastasis. Out of 34 patients included, the treatment demonstrated a 100% disease control rate in patients with PDL1≥1% expression, with a median time to symptom improvement of 12.5 days and significant intracranial and extracranial responses, suggesting that ABCP is feasible and comparably effective in this population as in patients without actionable mutations receiving upfront chemo-immunotherapy.

Minor Comments :

1. Authors have to indicate the Correlation of PD-L1 and EGFr in these patients (a correlation figure would be great)

Author Response

Thank you for your kind review and helpful comment.

We have updated the manuscript and added a correlation result. We have highlighted in green the new changes. The paragraph in question is in line 225 as follows:

"All patients with PDL1≥1% achieved disease control, and the presence of PDL1 expression had a positive correlation with radiological response (r=0.4191 (95%CI 0.01895-0.7036); r2 0.1757; p=0.0415), but the low r suggested that the absence of PDL1 expression did not guarantee a lack of response. There was no correlation between the presence of PDL1 expression and PFS or OS (data not shown), but there was a trend to better survival observed in patients with PDL1≥1% (HR 0.44; 95% CI 0.18-1.1; p=0.0896) (see Figure 2 A and B)."

Kind regards,

Carles Escriu

Reviewer 2 Report

Comments and Suggestions for Authors

The manuscript presents valuable insights into the real-life efficacy of ABCP in patients with lung cancer and brain metastases, contributing to the growing body of evidence supporting its use in this patient population.

Addressing the following points would further strengthen the clarity, comprehensibility, and impact of the study:

-Certain aspects of the Introduction, such as the critical analysis of existing evidence and the discussion of treatment paradigms, could potentially be condensed and moved to the discussion section. By doing so, the Introduction could focus more directly on setting up the rationale and objectives of the study, while the discussion section could delve deeper into the interpretation of results and implications for clinical practice.

-While the study provides valuable insights into the feasibility and efficacy of ABCP in real-life patients with brain metastases, it is essential to acknowledge more clearly the potential limitations, such as the retrospective nature of data collection and the inherent biases therein.

-Additionally, the authors should add/emphasize more in the conclusion: the need to focus further on high-need populations (untreated brain metastases and/or ALK translocations), the importance of personalized treatment approaches in patients with actionable mutations, avenues for future research, including prospective studies to validate these findings and exploring potential biomarkers predictive of treatment response.

Comments on the Quality of English Language

minor editing

Author Response

Thank you very much for taking the time to review this manuscript and the kind comments. We found the suggestions from the reviewer very helpful and we think they have made a very positive contribution to the manuscript. We have updated the manuscript with all the reviewers’ comments and highlighted the changes in green.

Point by point response:

-Certain aspects of the Introduction, such as the critical analysis of existing evidence and the discussion of treatment paradigms, could potentially be condensed and moved to the discussion section. By doing so, the Introduction could focus more directly on setting up the rationale and objectives of the study, while the discussion section could delve deeper into the interpretation of results and implications for clinical practice.

Reply: We have substantially shortened the introduction as advised and strengthened the rationale and objectives of the study. These changes can be apprecited in line 72 and 89 of the introduction. We have moved the two paragraphs from the introduction into the discussion, where we believe it flows better.

-While the study provides valuable insights into the feasibility and efficacy of ABCP in real-life patients with brain metastases, it is essential to acknowledge more clearly the potential limitations, such as the retrospective nature of data collection and the inherent biases therein.

Reply: Thank you for this comment. The limitations have been expanded in the discussion and can be found in line 340 of the updated manuscript.

-Additionally, the authors should add/emphasize more in the conclusion: the need to focus further on high-need populations (untreated brain metastases and/or ALK translocations), the importance of personalized treatment approaches in patients with actionable mutations, avenues for future research, including prospective studies to validate these findings and exploring potential biomarkers predictive of treatment response.

Reply: Thank you for the comment. We have added a new point in the discussion in line 556 titled "Clinical implications and future directions"

The manuscript has been reviewed by three native english speakers. 

Thank you again for your time.

Sincerely,

Carles Escriu